# Time Series Prediction in Industry 4.0: A Comprehensive Review and Prospects for Future Advancements

**Nataliia Kashpruk, Cezary Piskor-Ignatowicz and Jerzy Baranowski ***

Department of Automatic Control and Robotics, AGH University of Kraków, 30-059 Kraków, Poland;
nataliia@agh.edu.pl (N.K.); ignatow@agh.edu.pl (C.P.-I.)
* Correspondence: jb@agh.edu.pl

**Abstract:** Time series prediction stands at the forefront of the fourth industrial revolution (Industry 4.0), offering a crucial analytical tool for the vast data streams generated by modern industrial processes. This literature review systematically consolidates existing research on the predictive analysis of time series within the framework of Industry 4.0, illustrating its critical role in enhancing operational foresight and strategic planning. Tracing the evolution from the first to the fourth industrial revolution, the paper delineates how each phase has incrementally set the stage for today's data-centric manufacturing paradigms. It critically examines how emergent technologies such as the Internet of things (IoT), artificial intelligence (AI), cloud computing, and big data analytics converge in the context of Industry 4.0 to transform time series data into actionable insights. Specifically, the review explores applications in predictive maintenance, production optimization, sales forecasting, and anomaly detection, underscoring the transformative impact of accurate time series forecasting on industrial operations. The paper culminates in a call to action for the strategic dissemination and management of these technologies, proposing a pathway for leveraging time series prediction to drive societal and economic advancement. Serving as a foundational compendium, this article aims to inform and guide ongoing research and practice at the intersection of time series prediction and Industry 4.0.

**Keywords:** Industry 4.0; time series; forecasting; industrial revolutions; Internet of things (IoT); artificial intelligence (AI); big data; 5G; 3D; cloud computing

## 1. Introduction

In our rapidly developing world, technological advancements have been pivotal in shaping various aspects of human life. With the emergence of the COVID-19 pandemic, the urgency for digital transformation has been highlighted, showcasing the potential of technology in enabling not just remote work and learning, but also in managing unprecedented global health challenges. This has led to significant cost savings, enhanced efficiency of business processes, and in critical instances, life-saving interventions.

Time series prediction, an essential aspect of data analysis, has gained prominence within this technological pivot, particularly as it applies to Industry 4.0. This review examines the fourth industrial revolution, delving into the challenges and future research perspectives, with a keen focus on the role of time series analysis in technology integration and pandemic management. We aim to uncover the impact of key technological areas such as the Internet of things (IoT) and big data on industrial processes, which have historically spurred economic growth through significant production advancements.

The foundation of Industry 4.0 resonates with the attributes of time series data, pursuing objectives such as interconnection, information transparency, technical assistance, and decentralized decision making. These principles facilitate a network of seamless communication, allow for the comprehensive analysis of industrial data, and support informed decision making through aggregated data visualization. Moreover, Industry 4.0 champions

autonomous systems that operate with minimal human intervention, relying on the critical analysis of time series data.

In this literature review, we employed a systematic approach to explore the interplay between time series prediction and Industry 4.0. Our methodology entailed a comprehensive search of academic databases, including Scopus, Web of Science, IEEE Xplore, and Google Scholar, to collate pertinent literature published up to April 2023. Keywords such as "Industry 4.0", "time series prediction," "IoT," "AI," "cloud computing," "big data," and "predictive analytics" were used in various combinations to ensure a thorough retrieval of relevant documents.

Inclusion criteria were set to filter studies that specifically addressed the application of time series analysis within the context of Industry 4.0, including those that discussed predictive modeling, anomaly detection, and data-driven decision-making processes. References were also screened for relevance based on their abstracts and conclusions, leading to a curated list of literature that provides a holistic view of the current state of time series prediction technologies and their real-world industrial applications.

The selected literature was then reviewed, and data were extracted focusing on the objectives, methodologies, results, and conclusions of each study. This enabled us to identify common themes, technological trends, and gaps in the current body of knowledge. The review also included a critical assessment of the various technologies applied in time series prediction, comparing their impacts, challenges, and potential for future development within the landscape of Industry 4.0.

This methodical approach facilitated a structured narrative that tracks the progression of time series analysis from a niche analytical tool to a cornerstone of modern industrial strategy. By integrating this methodology into our introduction, we set the stage for a review that not only synthesizes existing research but also maps out the trajectory of technological evolution in the sphere of Industry 4.0.

This review embarks on a comprehensive exploration of the integral relationship between time series data and Industry 4.0, aiming to equip researchers and practitioners with insights that span across various disciplines. We navigate through the history of Industry 4.0, outline key technologies within its scope, and transition to a focused examination of multiple applications of time series data, including AI, IoT, cloud computing, predictive maintenance, and anomaly detection, culminating with the integration of pandemic management strategies.

The structure of the paper is as follows: We begin with the historical context of Industry 4.0, introduce the pivotal technologies underpinning it, and then delve into the specific applications of time series data within this industrial framework. The paper concludes with a summary of our findings, offering a perspective on the future of industry and technology's role in the response to global health crises.

## 2. Evolution of Industrial Revolutions: From Steam Power to Industry 4.0

The concept of the industrial revolution encompasses a series of transformative changes in production and industrial processes through the adoption of innovative technologies. These revolutions have helped to shape our civilization and have had profound effects on various aspects of society, economy, and culture. As we look back at history, we are reminded of the four major industrial revolutions that have significantly shaped the course of human progress.

Figure 1 depicts the evolution of industrial revolutions, each representing a pivotal shift in the way we produce goods, interact, and organize our lives. These revolutions have not only revolutionized industries but have also triggered great social transformations, altering the very fabric of human existence.

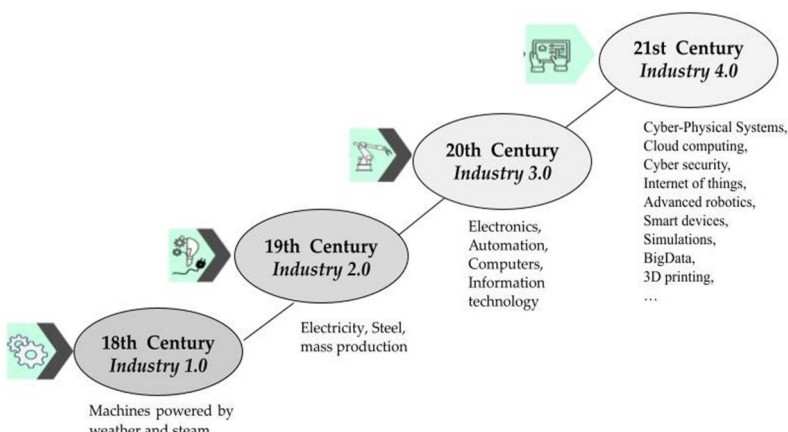

**Figure 1.** Evolution of industrial revolutions.

### 2.1. Industry 1.0—The First Industrial Revolution: The Era of Mechanical Production

The First Industrial Revolution began around 1760 with the introduction of water and steam engines, leading to the establishment of large factories as centers of production and industry. This revolutionary period saw the widespread adoption of steam engines, which powered various sectors, from agriculture to textile manufacturing. The development of steam power, alongside other machines, resulted in the transformation of society, marked by the rise of urbanization and the mechanization of manufacturing [1].

During this era, reliance on agriculture decreased, and societies shifted towards urban centers, driven by the increased use of steam engines and machine tools. Steamships and railroads further revolutionized transportation, making it faster and more efficient.

However, factory life during this period was arduous, with inexpensive and abundant unskilled workers forced to toil in hazardous conditions. Child labor was prevalent, and workers endured long, exhausting shifts. These harsh conditions persisted until the 20th century when the advancement of industrialization eventually led to the emergence of a middle class of skilled workers, contributing to the rapid growth of cities, industries, and economies.

### 2.2. Industry 2.0—The Second Industrial Revolution: The Era of Science and Mass Production

The Second Industrial Revolution occurred during the late 19th and early 20th centuries, characterized by rapid electrification, the expansion of railway and telegraph networks, and the introduction of key inventions such as gasoline engines, airplanes, and chemical fertilizers. These innovations enabled society to achieve greater speed and productivity.

Scientific principles were directly applied to the factory setting, most notably through the introduction of the assembly line, which revolutionized mass production. Henry Ford's introduction of the assembly line in the early 20th century enabled the mass production of the Ford Model T, a car with a gasoline engine, leading to increased accessibility to automobiles.

Urbanization continued to accelerate during this period, with workers leaving rural areas to work in factories in growing urban centers. The advancements in technology, such as electric lighting, radios, and telephones, also brought significant changes to people's lifestyles and communication [1].

### 2.3. Industry 3.0—The Third Industrial Revolution: The Digital Revolution

The Third Industrial Revolution emerged at the end of the 20th century, driven by the production and development of computers, computer networks, robotics in manufacturing, and the birth of the Internet and mobile phones. The digital revolution brought about significant changes in production processes, emphasizing automation, robotization, and the use of IT technologies [1].

Semiconductors, mainframes, personal computers, and the Internet played crucial roles in transforming analog devices into digital ones. This shift had a profound impact on industries, especially in global communications and energy. The automation of production and integrating supply chains around the world became prominent features of this era.

*2.4. Industry 4.0—The Fourth Industrial Revolution: It Starts Now*

Industry 4.0 was first mentioned in 2011 as a phenomenon of automation in production processes. This era is characterized by modern technologies speeding up the transformation of traditional industries. Artificial intelligence and advanced robotics [2] have ushered in the Fourth Industrial Revolution, leading to the automation of high-quality technical work previously performed by humans.

The Fourth Industrial Revolution marks a significant shift from the client-server model to ubiquitous mobility, integrating digital and physical environments through cyber-physical systems in production [3]. It combines various technologies such as the Internet of things, big data, cloud, advanced robotics, and artificial intelligence to create new opportunities for innovation and complete automation in industries, propelling them to new heights [1,4–7].

## 3. Key Technologies Shaping the Fourth Industrial Revolution

The Fourth Industrial Revolution is characterized and driven by a set of key technologies that are transforming industries and driving unprecedented advancements in production and automation. These transformative technologies are illustrated in Figure 2 [1,4,7,8]:

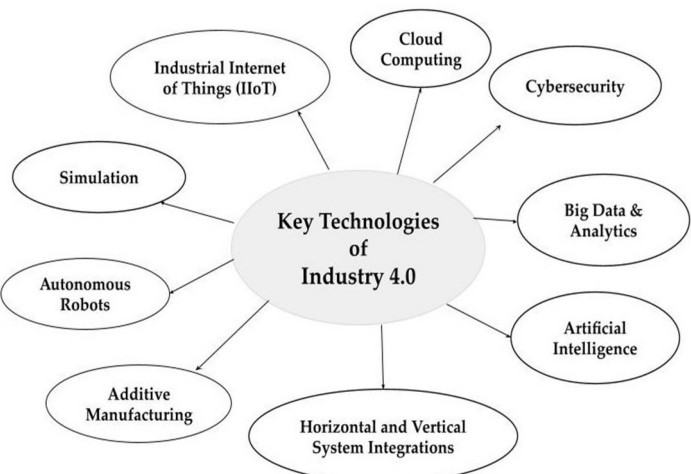

**Figure 2.** Key technologies of Industry 4.0.

Big Data and Analytics: This technology involves the collection, processing, and analysis of vast volumes of data to derive valuable insights, patterns, and trends. By employing advanced statistical algorithms and data visualization techniques, businesses can optimize their operations, enhance decision-making processes, and improve overall productivity. Practical examples of its implementation include predictive maintenance [9–11] in manufacturing, where sensors collect data from machines to anticipate potential breakdowns [12] and optimize maintenance schedules, leading to cost savings and minimized downtime [8].

Industrial Internet of Things (IIoT): The IIoT involves embedding smart sensors and interconnected devices in industrial equipment to gather real-time data and enable remote monitoring and control. In manufacturing, IIoT facilitates predictive maintenance [13,14], real-time production monitoring [15,16], and supply chain optimization. For instance, in a connected factory, sensors can monitor equipment health and usage patterns, leading to timely maintenance interventions and increased machine uptime [1].

Cloud Computing: Cloud-based services offer scalable, on-demand access to computing resources and storage. Manufacturing companies leverage cloud platforms to store

and process vast amounts of data generated from IoT devices, allowing them to analyze data [17] more efficiently and implement collaborative processes with stakeholders across different locations [1,6,18].

Artificial Intelligence (AI): AI is a transformative technology that plays a pivotal role in the Fourth Industrial Revolution, driving innovation and reshaping industries across various domains. AI refers to the development of computer systems capable of performing tasks that typically require human intelligence, such as problem solving, decision making, speech recognition, and language translation. The integration of AI with other key technologies of Industry 4.0, such as big data and analytics [2,19], IoT, and cloud computing, enhances its capabilities and potential for transformation in diverse industries. As AI continues to evolve, it holds the promise of revolutionizing how we live, work, and interact with technology in the digital era [1,8,20].

Simulation: Simulation technology employs computer modeling [21] and visualization to replicate real-world scenarios and predict outcomes. In the manufacturing sector, digital twins [22–24] are utilized to create virtual replicas of physical products or processes. Manufacturers can use these simulations to optimize designs, identify potential flaws, and streamline production processes, reducing development time and costs [25].

Cybersecurity: As the fourth industrial revolution brings increased connectivity, cybersecurity becomes paramount to safeguard against potential threats and data breaches. Industrial control systems and critical infrastructure need robust protection from cyberattacks. Advanced encryption, network security protocols, and intrusion detection systems are practical examples of cybersecurity measures implemented to ensure data integrity and continuity of operations [1].

Horizontal and Vertical System Integration: The integration of data networks horizontally and vertically within organizations enables seamless data sharing, real-time communication, and better coordination among different departments and hierarchical levels. In a smart factory setting, for example, the integration of data from production processes, supply chain, and maintenance allows for a comprehensive view of the entire manufacturing operation, leading to improved efficiency and resource allocation.

Autonomous Robots: Advanced robotic systems equipped with artificial intelligence, computer vision, and advanced sensors are capable of performing complex tasks [2,26–28] with minimal human intervention. These robots find practical applications in various industries, such as logistics and warehousing, where they can efficiently handle order fulfillment, inventory management, and material handling, significantly enhancing operational efficiency [6].

These technologies are applied in various ways across industries, including manufacturing, healthcare, logistics, and more. They are revolutionizing traditional processes, optimizing efficiency, and driving innovation to usher in the Fourth Industrial Revolution. It should be noted that the advancement of Industry 4.0 technologies is characterized by constant dynamism and evolution. As a result, the key technologies may differ based on the unique needs and demands of various industries and enterprises.

## 4. Advancements in Industry 4.0 and Pandemic Management

The advancements in Industry 4.0 have revolutionized manufacturing processes and decision making in various industries. The integration of digital technologies, artificial intelligence [29,30], and data-driven approaches has paved the way for smart factories and efficient production systems. Additionally, the COVID-19 pandemic has accelerated the adoption of Industry 4.0 technologies in pandemic management scenarios. This chapter presents a comprehensive review of scientific papers that highlight the applications of artificial intelligence, machine learning, IoT, cloud computing, time series analysis [18,31–34], and predictive maintenance [35] in the context of Industry 4.0. The papers explore innovative methodologies, algorithms, and solutions that contribute to enhancing efficiency, optimizing processes, and addressing challenges in modern manufacturing.

### 4.1. Artificial Intelligence and Machine Learning

Advancements in artificial intelligence (AI) and machine learning (ML) have been at the forefront of transforming industries, and Industry 4.0 represents a significant step in this direction. Industry 4.0, also known as the Fourth Industrial Revolution, brings together digital technologies, IoT, and AI to revolutionize manufacturing processes, enhance efficiency, and create smart factories. In this context, AI and ML play a vital role in optimizing various aspects of Industry 4.0, ranging from combating the COVID-19 pandemic [36] to forecasting in production facilities [6,30,32,37–39].

In a study by Ahmad [40], the potential of Industry 4.0 in combating the COVID-19 pandemic is explored. The authors focus on using deep convolutional neural networks (CNN) to analyze clinical radiological images, such as X-rays and computed tomography (CT), for early diagnosis of COVID-19 patients. The proposed CNN model achieves high classification accuracy, enabling automated patient analysis with minimal human contact.

Balamurugan et al. [41] investigate the applications of artificial intelligence and machine learning in the Manufacturing 4.0 sector. They highlight how IoT, artificial intelligence, cyber-physical systems, and machine learning contribute to a revolutionary change in the industry, allowing the creation of smart factories and process automation.

Bécue, Praça, and Gama [42] discuss the opportunities and threats related to adopting artificial intelligence in the manufacturing sector. The paper focuses on offensive and defensive uses of AI technology in Industry 4.0, emphasizing security principles and detection techniques for operational technology (OT) in manufacturing systems.

Chen and Wang [43] propose a hybrid approach called BD-I4, which combines big data analytics and Industry 4.0 technology, to enhance cycle time range projections for manufacturing jobs. By leveraging big data analytics with artificial intelligence and machine learning, the BD-I4 approach improves forecasting accuracy through collaboration among multiple experts.

Diez-Olivan, et al. [19] discuss the concept of "smartization" in manufacturing industries as part of Industry 4.0. They explore the use of data fusion and machine learning strategies to extract relevant knowledge from monitored assets and predict abnormal behaviors in industrial machinery, tools, and processes, contributing to efficient production processes.

Jácome-Galarza et al. [44] review efficient deep learning architectures for time series data produced by IoT sensors, with applications in smart cities, industry 4.0, sustainable agriculture, and robotics. The paper highlights the capabilities of long short-term memory (LSTM), convolutional neural networks (CNN), recurrent neural networks (RNN), and stacked LSTM autoencoders in time series prediction.

Javaheri et al. [45] conduct a comprehensive survey of DDoS attacks in the Industry 4.0 era. They recommend effective defensive strategies, particularly focusing on fuzzy-logic-based detection methods, to mitigate cybersecurity threats and improve business sustainability in the face of distributed denial of service (DDoS) attacks.

Tang et al. [46] propose GRN, an interpretable multivariate time series anomaly detection method, based on neural graph networks and gated recurrent units (GRU). GRN improves detection performance by automatically learning correlations between sensors and capturing long-term and short-term dependencies in industrial control systems.

Zeiser et al. [47] evaluate deep unsupervised anomaly detection methods for industrial production. Their data-centric approach, involving domain randomization and synthetic data, utilizes a combination of WGAN and encoder CNN to detect anomalies in online inspection, proving the effectiveness of clustering methods in anomaly differentiation.

In summary, the group of scientific papers presents diverse applications of artificial intelligence and machine learning in the context of Industry 4.0. The research focuses on utilizing technologies such as deep neural networks, big data analytics, and machine learning [48,49] to improve production efficiency, optimize processes, and predict anomalies in industrial systems. The adoption of AI and ML enables the creation of smart factories, provides process automation, and supports the fight against the COVID-19 pandemic.

These studies contribute valuable insights to the evolving field of Industry 4.0, opening new possibilities and perspectives for industry and beyond. A comparative summary of this section can be found in Table 1.

**Table 1.** Comparative summary of research on AI and ML integration in Industry 4.0 applications.

| Study | Focus | AI/ML Technologies |
|-------|-------|--------------------|
| [40] | COVID-19 diagnosis using radiological images | Deep convolutional neural networks (CNN) |
| [41] | Smart factories and process automation | IoT, artificial intelligence (AI), cyber-physical systems, machine learning (ML) |
| [42] | AI in manufacturing for offensive and defensive uses | Security principles, detection techniques |
| [43] | Forecasting for manufacturing jobs | BD-I4 (big data and Industry 4.0) |
| [50] | "Smartization" of manufacturing industries | Data fusion, ML strategies |
| [44] | Deep learning architectures for time series data | Long short-term memory (LSTM), convolutional neural networks (CNN), recurrent neural networks (RNN), stacked LSTM autoencoders |
| [45] | DDoS attacks in Industry 4.0 | Fuzzy-logic-based detection methods |
| [46] | Time series anomaly detection | Neural graph networks, gated recurrent units (GRU) |
| [47] | Unsupervised anomaly detection in production | WaterGAN (WGAN), encoder CNN |

### 4.2. Internet of Things (IoT)

The Internet of things (IoT) has emerged as a revolutionary concept that connects physical devices and objects to the internet, enabling them to collect and exchange data autonomously. In recent years, IoT has found extensive applications in various fields, with Industry 4.0 being one of its prominent domains. Industry 4.0 represents the Fourth Industrial Revolution, characterized by the integration of advanced technologies into manufacturing and production processes to create smart, efficient, and interconnected factories.

In this group of papers, we explore several key aspects of IoT in the context of Industry 4.0. The authors delve into innovative approaches, systems, and architectures to enhance the implementation and utilization of IoT technologies [51] in transforming industries and optimizing various processes. The papers address challenges faced by small- to medium-sized factories, data engineers, and city managers in deploying and utilizing IoT solutions.

Kuo et al. [52] propose an automatic machine status prediction system for Industry 4.0. Recognizing the financial constraints of smaller factories, they introduce the use of inexpensive add-on triaxial sensors for machinery monitoring. This cost-effective approach, coupled with a neural network and dimensionality reduction method, allows for automatic analysis of collected data, making Industry 4.0 more accessible to smaller enterprises.

Villalobos et al. [53] present the Interface for Time Series Reduction System (I4TSRS), which assists data engineers in reducing the dimensionality of time-series data for Industry 4.0 scenarios. The system recommends techniques to obtain reduced representations of industrial time-series data while preserving their main characteristics. This reduction in data size lowers storage and transmission costs without compromising data exploitation in various processes.

Same authors, in their later work, Villalobos et al. [54] propose a three-level hierarchical architecture for efficient storage of Industry 4.0 time-series data in cloud environments. By categorizing and storing data at different levels, the architecture optimizes storage resources, allowing for cost-effective and scalable data management.

Chen et al. [55] explore data dissemination in the context of Industry 4.0 applications within the Internet of vehicles (IoV). Their novel data dissemination scheme relies on short-term traffic prediction using a deep learning network. This scheme empowers city managers to regulate traffic flow efficiently and fosters the development of various Industry 4.0 applications in IoV, such as smart logistics and autonomous driving.

Finally, Enes et al. [56] introduce a pipeline architecture for feature-based unsupervised clustering using multivariate time series from high-performance computing (HPC) jobs. The pipeline's scalability and unsupervised nature make it suitable for a wide range of industrial and research applications, enabling meaningful clustering of large-scale data sets.

Together, these papers shed light on the potential of IoT in Industry 4.0 and offer practical solutions to address challenges in implementing IoT technologies in various industrial contexts. The research presented in this group contributes to advancing IoT's transformative capabilities, revolutionizing industries, and enhancing operational efficiency. Its overview is provided in Table 2.

**Table 2.** Overview of IoT innovations and challenges in the advancement of Industry 4.0.

| Study | Innovation or Challenge Addressed | IoT Application and Outcome |
|---|---|---|
| [51] | Implementation and utilization of IoT | Enhanced industry transformation and process optimization |
| [52] | Machine status prediction with financial constraints for small factories | Inexpensive triaxial sensors and neural networks for accessible Industry 4.0 |
| [53] | Time series data dimensionality for data engineers | I4TSRS for efficient data storage and transmission in industrial processes |
| [54] | Hierarchical storage architecture for cloud environments | Cost-effective data management for Industry 4.0 time series data |
| [55] | Data dissemination in IoV for city managers | Short-term traffic prediction and smart logistics management |
| [56] | Feature-based unsupervised clustering for HPC jobs | Scalable clustering of large-scale data sets for industrial and research applications |

*4.3. Cloud Computing*

This group centers on cloud computing, a pivotal element within the context of Industry 4.0, also known as the Fourth Industrial Revolution. Industry 4.0 signifies a paradigm shift in industrial transformation, leveraging advanced information technologies, automation, data analytics [17], and artificial intelligence to revolutionize manufacturing processes. Cloud computing plays a critical role in this context, enabling flexible, scalable, and cost-effective storage, processing, and sharing of data and applications.

Villalobos et al. [54] propose a novel three-level hierarchical architecture aimed at achieving efficient storage of Industry 4.0 time series data within cloud environments. The growing interest in monitoring and analyzing industrial systems has resulted in significant data generation, leading to considerable costs associated with data storage. To address this issue, the authors present a hierarchical storage scheme, wherein new raw time series are stored on electronic non-volatile storage (solid-state drives—SSDs) for short periods, recent time series are stored on magnetic hard disk drives (HDDs) for medium periods, and a reduced representation of time series, obtained through time series reduction techniques, is stored in HDDs for longer durations. The proposed architecture mitigates storage expenses while preserving the usability of data for future analysis.

Moosavi, Bakhshi, and Martek [57] conduct a scientometric analysis to explore the application of Industry 4.0 technologies in pandemic management, particularly during the COVID-19 outbreak. The study highlights key technologies associated with Industry 4.0, including the Internet of things (IoT), artificial intelligence (AI), cloud computing, machine learning, security, big data, blockchain, deep learning, digitalization, and cyber-physical systems (CPS). The authors subsequently present a case study exemplifying the application of Industry 4.0 technologies in managing the COVID-19 pandemic. The study concludes that the COVID-19 crisis has acted as a catalyst in accelerating the digital transformation towards Industry 4.0 and anticipates the widespread use of these technologies in future pandemic management scenarios.

Chen and Wang [43] propose a novel hybrid approach, termed big data analytics and Industry 4.0 (BD-I4), to enhance the precision of cycle time range projections for factory jobs. The BD-I4 approach synergistically combines big data analytics, specifically deep learning, with principles of Industry 4.0, particularly artificial intelligence. In this approach, individual experts construct fuzzy deep neural networks to project cycle time ranges for specific jobs. These projected cycle times are subsequently aggregated using the fuzzy weighted intersection operator to consider unequal authority levels. The experimental application of the BD-I4 approach to real-world scenarios demonstrates a remarkable improvement in projection precision, underscoring the efficacy of collaborative expertise for enhanced forecasting.

Zalte-Gaikwad et al. [58] present a comprehensive book exploring the convergence of big data with cloud computing for Industry 4.0 applications. The book provides insights into adapting and implementing big data technologies in diverse cloud domains, with a specific focus on their integration within Industry 4.0. The authors emphasize the development of adaptive, robust, scalable, and reliable applications capable of addressing day-to-day challenges in manufacturing and production as part of Industry 4.0. Utilizing case studies and expert contributions from leading big data professionals worldwide, the book addresses key challenges, advancements, and issues in the context of big data and cloud computing for Industry 4.0.

Grigoriou et al. [59] emphasize the significance of cloud computing as a fundamental enabler for smart factories and smart warehouses within the realm of Industry 4.0. They underscore the paradigm shift towards cloud-based architectures, offering easier accessibility and cost-effectiveness for larger robotic systems. The authors present an overview of cloud technology, detailing its characteristics, service models, and deployment models. Additionally, they elaborate the indispensable role of cloud computing in Industry 4.0, its implications, and the challenges faced in comparison to traditional systems.

Gautam [60] conducts a comparative analysis between edge computing and cloud computing technologies for Industry 4.0 perspectives. The rapid growth of the Internet of things (IoT) and connected devices has led to substantial data generation, necessitating efficient analysis and processing. Cloud computing technology has traditionally served these purposes; however, a trend towards employing computational technologies closer to devices has emerged, termed edge computing. This article critically examines the benefits and drawbacks of both technologies and their potential applications within the context of Industry 4.0.

Pandey et al. [61] provide, a comprehensive overview of Industry 4.0, intelligent manufacturing, IoT, and cloud computing. The chapter traces the evolution of Industry 4.0 and its impact on digital transformation in manufacturing and related industries. It delves into the ecosystem of intelligent sensors, devices, and applications, highlighting their role in increasing productivity and streamlining business operations. Furthermore, the integration of manufacturing operational technology with information technology through data science, machine learning, and artificial intelligence is explored. The authors also emphasize the crucial role of cloud computing in supporting IoT at a large scale and economically. Security controls and best practices in achieving smart manufacturing realization conclude the chapter.

In summary, the papers in this group provide valuable insights into the crucial role of cloud computing within the context of Industry 4.0. As the Fourth Industrial Revolution continues to transform manufacturing processes, cloud computing emerges as a pivotal element, offering flexible, scalable, and cost-effective solutions for data storage, processing, and sharing. The rigorous research presented in this group significantly contributes to the understanding and practical implementation of cloud computing technologies within the paradigm of Industry 4.0. Key contributions of cloud computing are highlighted in Table 3.

**Table 3.** Key contributions of cloud computing to Industry 4.0: applications, models, and impacts.

| Study | Cloud Computing Application | Outcome/Contribution |
|---|---|---|
| [54] | Hierarchical storage architecture | Efficient storage, reduced costs for time series data |
| [57] | Scientometric analysis and case study on COVID-19 management | Highlighting acceleration of digital transformation towards Industry 4.0 |
| [43] | BD-I4 approach for cycle time projections | Improved forecasting accuracy with collaborative expertise |
| [58] | Convergence of big data and cloud computing for Industry 4.0 | Development of scalable applications for Industry 4.0 challenges |
| [59] | Cloud-based architectures for smart factories | Enhanced accessibility and cost-effectiveness |
| [60] | Comparative analysis of edge and cloud computing technologies | Examination of benefits and drawbacks for Industry 4.0 applications |
| [61] | Overview of IoT and cloud computing in intelligent manufacturing | Emphasizes cloud computing's support for IoT and smart manufacturing |

### 4.4. Predictive Maintenance and Prognosis

Predictive maintenance [24,62–65] and prognosis play a crucial role in the context of Industry 4.0, facilitating the transition from scheduled-based processes to smart, reactive ones. This review explores several scholarly papers that focus on predictive maintenance methodologies and their applications [49,66–69] in the context of Industry 4.0.

Ruiz-Sarmiento et al. [70] present an Industry-4.0-based approach for the health assessment of critical assets in the stainless steel industry. The authors propose a predictive model based on a Bayesian filter from the machine learning field to estimate and predict the gradual degradation of machinery, specifically the drums within the heating coilers of Steckel mills. By fusing expert knowledge with real-time information, the model enables informed decisions regarding maintenance operations, contributing to the Industry 4.0 era.

Akkaya et al. [71] delve into emerging trends and strategies for Industry 4.0 during and beyond the COVID-19 pandemic. The book provides multidisciplinary research and insights on advancing technologies and new strategies in business and administrative settings. The authors reflect upon the challenges and opportunities brought about by the pandemic, and how it has accelerated the adoption of Industry 4.0 technologies.

Züfle et al. [72] propose a generic end-to-end predictive maintenance methodology for estimating the time-to-failure of industrial machines. The methodology includes feature extraction based on various sensor data types, feature transformation and selection techniques, and the training of multiple machine learning classification models. The methodology is evaluated through a real-world case study, demonstrating its effectiveness for time-to-failure prediction.

Leander et al. [73] discuss access control strategies for Industry 4.0 manufacturing systems, emphasizing the need for fine-grained access control policies to mitigate emerging cybersecurity threats. The authors evaluate several access control strategies in a simulation experiment, considering various attack scenarios, and outline a method for automatic policy generation based on engineering data.

Sang et al. [74] propose PMMI 4.0, a predictive maintenance model for Industry 4.0, which utilizes a data-driven LSTM model for remaining useful life (RUL) estimation. The model is integrated with a solution called PMS4MMC to support an optimized maintenance schedule plan for multiple machine components. The effectiveness of the proposed solution is demonstrated through a real-world industrial case.

In their paper, Torim et al. [75] present a flexible architecture for predictive maintenance systems, incorporating expert knowledge and matrix-profile anomaly detection for advanced failure prediction. The system aims to enable timely corrective actions and prevent material and environmental damages in the context of Industry 4.0. The integration of continuous monitoring and healthiness checks allows for accurate predictions of equipment failures and facilitates proactive decision making by the maintenance team. The proposed conceptual model showcases the potential of combining expert knowledge with anomaly detection techniques, yielding promising results for the implementation of predictive maintenance in various industrial sectors.

Gautam et al. [76] discuss the application of machine learning and IIoT (industrial Internet of things) for predictive maintenance. The paper explores the potential of ML techniques to analyze data from IIoT tools, such as smart sensors, to predict equipment performance and health. A case study on data from a heat exchanger showcases the advantages of deep learning LSTM models in time series forecasting for predictive maintenance.

In conclusion, the papers in this group highlight the importance of predictive maintenance and prognosis in Industry 4.0. They provide valuable insights into the development of predictive models, the utilization of advanced technologies, and the implementation of effective strategies to enhance maintenance operations and optimize industrial processes in the context of Industry 4.0. This is summarized in Table 4.

**Table 4.** Advancements in predictive maintenance and prognosis strategies within Industry 4.0.

| Study | Predictive Maintenance Application | Outcome/Contribution |
|---|---|---|
| [70] | Health assessment of critical assets | Predictive model based on Bayesian filter |
| [71] | Trends and strategies during COVID-19 | Insights on accelerated adoption of Industry 4.0 technologies |
| [72] | Time-to-failure estimation methodology | Effective for machine failure prediction |
| [73] | Access control strategies for manufacturing | Method for automatic policy generation |
| [74] | PMMI 4.0: predictive maintenance model | Data-driven model for RUL estimation |
| [75] | Flexible architecture for predictive maintenance | Advanced failure prediction and decision making |
| [76] | ML and IIoT for predictive maintenance | Deep learning LSTM model for equipment performance prediction |

### 4.5. Time Series Analysis and Anomaly Detection

The field of time series analysis [77] and anomaly detection [15,35,78–82] has gained significant attention in recent years, especially in the context of Industry 4.0. This review focuses on a collection of scholarly papers that present innovative methodologies, algorithms, and techniques aimed at efficiently monitoring, predicting [31,83,84], and detecting anomalies in time series data from various industrial applications.

Zhang et al. [85] conducted a study on predicting the occurrence of the Fourth Industrial Revolution using time series analysis. They established an industrial revolution time prediction function based on quadratic functions and a time prediction model based on the gray GM (1,1) model. By analyzing the occurrence time sequences of the previous three industrial revolutions, the authors forecast that the fourth industrial revolution is likely to take place around 2055. This forthcoming revolution is expected to be characterized by technological advancements such as the Internet of things, big data, cloud computing, and intelligent manufacturing.

Authors of this review [86] propose a predictive model for system degradation using Bayesian time series models. The authors emphasize the importance of efficient maintenance in industrial equipment, and they address the limitations of deterministic prediction models that fail to consider the uncertainty inherent in degradation measurements. To overcome this, they introduce the use of time series models obtained through the Facebook Prophet algorithm, allowing for the prediction of degradation evolution in turbomachinery. The model is validated using data from large-scale industrial centrifugal compressors, and it demonstrates promising results with well-covered confidence intervals.

López-Blanco et al. [87] focused on improving the quality of life and ecosystem services in smart cities through time series forecasting. They recognized that enhancing the perception of quality of life requires action on the green infrastructure of cities. The researchers utilized state-of-the-art methodologies, including IoT technology, big data, and artificial intelligence, to collect environmental data and performed time series prediction processes using generalized additive models (GAM). Unlike LSTM and ARIMA models, GAM proved to be more efficient in handling seasonality issues in the data. The work enabled informed decision making by local authorities regarding the implementation of urban actions, ensuring a positive impact on citizens and the environment.

Dimoudis et al. [88] contribute to the field with an adaptive window rolling median methodology for time series anomaly detection. Given the rise of Industry 4.0 and the massive amounts of data collected from sensors, anomaly detection has become a crucial task. Existing anomaly detection methods are diverse due to the variety of anomalies encountered, and there is no universal approach. To address this challenge, the authors propose a novel anomaly detection algorithm that utilizes a rolling median with an adaptive sliding window. The window adapts based on two methods, F1-based and T-test, which aim to ensure an upward trend in F1 score and recognize trends in time series, respectively. The proposed algorithm is evaluated on benchmark data sets and real industrial machinery sensor observations, showing superior performance compared to an ensemble of seven existing models.

Javaheri et al. [45] present a comprehensive survey focusing on fuzzy-logic-based distributed denial of service (DDoS) attacks and network traffic anomaly detection methods. In the context of Industry 4.0, where business continuity is of utmost importance, cybersecurity challenges have become a major concern for information-technology-driven organizations and companies. DDoS attacks have proven to be particularly detrimental, leading to system failures, service disruptions, and significant financial losses. The paper offers a systematic overview of various DDoS attacks and proposes a hierarchical classification. Moreover, it conducts in-depth comparisons of studies published in reputed venues in this area. The authors recommend the adoption of fuzzy-based detection methods to mitigate DDoS threats effectively and address the gaps in current intrusion detection systems and related works.

Lugaresi et al. [23] introduce an online validation method for simulation-based digital twins, a prominent feature of Industry 4.0. As the digitization of manufacturing plants has increased, digital twins have become crucial for aligning physical systems with their corresponding digital models throughout the system's lifecycle. Traditional validation techniques are often limited by restrictive assumptions and the requirement of large data sets. To overcome these challenges, the authors propose a novel method that treats shop-floor data as sequences and measures the similarity between the data streams from the physical system and its corresponding digital model. The proposed method is validated through offline experiments and applied within a digital twin architecture for a lab-scale manufacturing system.

Stahmann and Rieger [89] contribute to the evaluation and selection of real-time anomaly detection algorithms applied in Industry 4.0 settings. With the advent of flexible, combinable production machines in Industry 4.0, timely and automated anomaly recognition has become crucial to support efficiency. However, existing evaluation mechanisms for real-time anomaly detection algorithms are not well-suited to the specific context of Industry 4.0. To address this gap, the authors develop an Industry-4.0-specific benchmark for real-time anomaly detection algorithms. This benchmark considers key design principles such as timeliness, threshold setting, and qualitative classification. The proposed benchmark allows for the ranking of algorithms based on context-specific input parameters for real-time anomaly detection in production datasets. The authors demonstrate the application of the benchmark through two case studies.

In conclusion, the papers in this group contribute significantly to the advancement of time series analysis and anomaly detection in the context of Industry 4.0. These scholarly works provide valuable insights, methodologies, and algorithms that enhance the understanding and practical implementation of time series analysis [66,90–92], predictive maintenance, anomaly detection, and cybersecurity measures in industrial settings. The research presented in this group has the potential to revolutionize maintenance practices, improve production efficiency, and ensure the reliability and security of industrial systems within the Industry 4.0 paradigm. The potential impact is summarized in Table 5.

**Table 5.** Innovations in time series analysis and anomaly detection techniques for Industry 4.0 applications.

| Study | Focus of Research | Impact on Industrial Applications |
|-------|-------------------|-----------------------------------|
| [66] | Prediction of the Fourth Industrial Revolution | Forecasting the occurrence and character of the Fourth Industrial Revolution |
| [86] | Predictive model for system degradation | Bayesian time series models for turbomachinery |
| [87] | Quality of life in smart cities | Time series forecasting for urban planning |
| [88] | Anomaly detection in sensor data | Adaptive window rolling median methodology |
| [45] | DDoS attacks and network traffic | Fuzzy-logic-based anomaly detection methods |
| [23] | Validation of digital twins | Online validation method for digital twins |
| [89] | Real-time anomaly detection algorithms | Benchmark for real-time anomaly detection |

*4.6. Advancements in Industry 4.0 and Pandemic Management*

This group comprises diverse articles exploring various aspects related to Industry 4.0 [93] and its impact on decision making, manufacturing, technology integration, and pandemic management. The articles delve into advanced data analysis techniques, emerging technologies, and the utilization of artificial intelligence to optimize processes, enhance decision making, and tackle challenges arising from the COVID-19 pandemic. The integration of digital technologies and data-driven approaches has become a pivotal driver in shaping the future of industries, paving the way for increased efficiency, resilience, and innovation.

Para et al. [94] conducted a study on decision making in Industry 4.0 scenarios, focusing on imbalanced data classification. They proposed a practical methodology for data analysis in industrial environments, emphasizing the importance of hypothesis generation dynamics among multidisciplinary experts prior to data capture. The study utilized a real industrial case study to address defect reduction, resulting in an imbalanced data classification problem. Extensive benchmarking of supervised learning algorithms and balancing preprocessing techniques were performed to accurately predict defective parts, thereby achieving a higher manufacturing quality.

Liu [95] introduced a big-data-driven macroeconomic forecasting model and analyzed psychological decision behavior in the context of Industry 4.0. The paper emphasized the significance of big data technology in providing multilevel, diversified, and comprehensive information for economic forecasting. The proposed model demonstrated higher accuracy and timeliness in predicting the Consumer Price Index (CPI) value in China, as well as analyzing the growth rate of industrial value added, indicating advantages over traditional forecasting methods. Additionally, the study explored the development and advantages of big-data-driven psychological decision-making behavior analysis, offering new insights for future research in this area.

Wichmann et al. [96] presented a comprehensive literature review on Industry 4.0, which represents the convergence of technologies in intelligent manufacturing, integrating information and communication technologies with distributed control systems. The paper identified reoccurring themes and trends in Industry 4.0 and discussed their expected effects on future manufacturing. It highlighted central characteristics, challenges, and opportunities, offering valuable support for developing actionable strategies in the context of Industry 4.0.

Erboz [97] defined Industry 4.0 as a strategic approach to digitalization in manufacturing, particularly focusing on the creation of highly automated industries through human-machine interaction. The paper analyzed the main drivers of technological advances in Industry 4.0, such as greater integration, optimal business solutions, and enhanced organizational communication. The work provided insights into future business models and the potential impact of cyber-physical systems and the Internet of things on Industry 4.0.

Majstorovic et al. [98] explored the integration of enterprise resource planning (ERP) in the context of Industry 4.0. The study highlighted the importance of IoT, big data analytics, AI, and cloud computing in enabling new manufacturing systems automation models. By integrating these elements, ERP systems can be enhanced to effectively support Industry 4.0, enabling improved planning and control from the resource to supply chain levels.

Moosavi et al. [57] conducted a scientometric analysis to present a systematic literature review on Industry 4.0, focusing on key technologies such as the Internet of things, artificial intelligence, cloud computing, and big data. The paper explored the transformation towards Industry 4.0 accelerated by the COVID-19 pandemic. Additionally, it discussed a case study on using Industry 4.0 technologies in pandemic management, revealing the potential for future application of these digital transformation technologies in pandemic scenarios.

Green et al. [99] conducted an interrupted time series study to analyze the impact of COVID-19-related misinformation on Twitter during the UK national lockdown announcement. The research aimed to understand how misinformation influences official information sharing during major government announcements amidst the COVID-19 pan-

demic. The paper identified distinct clusters of misinformation topics and observed an increase in COVID-19-related bot activity post-announcement.

In conclusion, the articles in the last group provide valuable insights into the dynamic landscape of Industry 4.0 and its multifaceted applications. Researchers and practitioners are increasingly embracing data science and advanced technologies to unlock the potential of modern manufacturing. Decision-making processes are undergoing significant transformations, with data analytics playing a central role in unveiling complex patterns and facilitating evidence-based strategies. Furthermore, the articles highlight the importance of exploring diverse technologies such as the Internet of things, artificial intelligence, and big data analytics to improve forecasting accuracy [5,100,101], enhance macroeconomic analyses, and address misinformation challenges during the pandemic. These technologies enable real-time monitoring, predictive modeling, and data-driven decision making, empowering industries to adapt swiftly to evolving conditions. The implications of Industry 4.0 extend beyond the factory floor, as seen in the pandemic management case study. The digital transformation facilitated by Industry 4.0 technologies has expedited the response to global health crises, demonstrating the potential for innovation and collaboration in addressing pressing challenges. In conclusion, the articles presented in this group shed light on the transformative power of Industry 4.0 and its role in shaping decision-making, manufacturing processes, and pandemic management. The integration of data-driven methodologies and advanced technologies enables industries to optimize operations, improve forecasting accuracy, and respond effectively to dynamic global situations. As we continue to explore the possibilities of Industry 4.0, it becomes evident that its potential for positive impact on various domains is vast and continues to expand. This is summarized in Table 6.

**Table 6.** Role of Industry 4.0 in enhancing decision making and managing pandemics: a summary of advancements and outcomes.

| Study | Focus of Advancement | Outcomes and Implications |
|---|---|---|
| [94] | Decision making in Industry 4.0 | Improved data classification for manufacturing quality |
| [95] | Big-data-driven macroeconomic forecasting and decision behavior | Enhanced economic forecasting and analysis of psychological decision behavior |
| [96] | Review of Industry 4.0 technologies | Identified trends and impact on manufacturing |
| [97] | Digitalization in manufacturing | Insights into future business models and impact of CPS |
| [98] | ERP integration in Industry 4.0 | Enhanced ERP systems for improved planning and control |
| [57] | Scientometric analysis on Industry 4.0 | Insights into digital transformation accelerated by COVID-19 |
| [99] | Impact of COVID-19 misinformation | Studied misinformation effects on public information sharing |

In conclusion, the reviewed scientific papers offer valuable insights into the transformative capabilities of Industry 4.0 and its impact on various aspects of manufacturing and pandemic management. The integration of advanced technologies such as AI, IoT, and cloud computing has enabled the creation of smart factories and the development of predictive maintenance strategies. Time series analysis and anomaly detection methodologies [102–104] have proven effective in monitoring industrial systems and detecting abnormal behaviors. Additionally, Industry 4.0 technologies have played a crucial role in pandemic management, offering data-driven approaches to address challenges during the COVID-19 crisis. The research presented in this chapter contributes significantly to the ongoing advancements in Industry 4.0, opening new possibilities and perspectives for the industry and beyond.

### 4.7. Comparative Analysis of AI, IoT, and Cloud Computing in Industry 4.0

The individual and collective impacts of Artificial Intelligence (AI), the Internet of things (IoT), and cloud computing technologies serve as the backbone of Industry 4.0, each contributing uniquely to the paradigm shift in industrial processes. This section presents a comparative analysis of these technologies, emphasizing their distinct and synergistic

effects on process optimization and predictive modeling, while also discussing the specific challenges associated with each.

### 4.7.1. Artificial Intelligence (AI)

AI's impact on Industry 4.0 is predominantly seen in its ability to facilitate complex decision-making processes and predictive maintenance. Through machine learning algorithms and neural networks, AI processes large volumes of data for failure prediction, quality control, and production optimization. However, the challenges lie in the need for high-quality data, the complexity of model interpretation, and the ongoing requirement for human oversight to contextualize AI outcomes.

### 4.7.2. Internet of Things (IoT)

IoT technologies provide real-time monitoring and control, enabling a network of interconnected devices to communicate and make decentralized decisions. This leads to significant improvements in operational efficiency and asset management. IoT's challenges include ensuring data security, managing the vast quantity of data from sensors, and integrating disparate systems across different standards and protocols.

### 4.7.3. Cloud Computing

Cloud computing offers scalable resources for storage and computation, supporting the vast data requirements of Industry 4.0. It allows for the flexible deployment of applications and analytics tools, which are pivotal for predictive modeling. However, cloud technologies face challenges such as latency in data access, data privacy concerns, and reliance on continuous internet connectivity.

### 4.7.4. Synergistic Effects

When combined, AI, IoT, and cloud computing have a multiplicative effect on process optimization and predictive modeling. AI's predictive capabilities, IoT's real-time data collection, and cloud computing's processing power create a cohesive framework that can dramatically increase the efficiency and agility of industrial operations. This integrated approach enables advanced analytics, such as predictive maintenance and optimization algorithms, to run efficiently at scale.

### 4.7.5. Specific Challenges

Each technology faces unique challenges when integrated into Industry 4.0's ecosystem. AI requires continuous improvement and adaptation to changing data patterns. IoT must ensure seamless integration and communication between devices. Cloud computing has to guarantee high availability and robust security measures. Addressing these challenges is crucial for the successful implementation and operation of Industry 4.0 technologies.

In conclusion, while AI, IoT, and cloud computing each bring distinct advantages to Industry 4.0, their interplay is essential for realizing the full potential of smart manufacturing. The continuous evolution of these technologies, coupled with strategic management and mitigation of their respective challenges, will dictate the trajectory of Industry 4.0's success.

## 5. Conclusions and Future Directions

The exploration of time series analysis within the expansive context of Industry 4.0 reveals a compelling narrative of technological synergy. This review has highlighted the pivotal role of time series prediction in catalyzing the efficiency and innovation that characterize the Fourth Industrial Revolution. The integration of artificial intelligence, the Internet of things (IoT), cloud computing, and big data analytics, centered around time series analysis, underscores the strategic importance of predictive insights in industrial automation and decision-making processes. The onset of the COVID-19 pandemic has not only accelerated the digital shift but also underscored the critical role of predictive modeling in managing complex, dynamic crises.

In the era of Industry 4.0, the application of time series analysis extends from optimizing manufacturing workflows to enabling predictive maintenance. The predictive power of AI, realized through the analysis of sequential data, is enhancing the foresight of industrial processes. Meanwhile, IoT contributes a continuous stream of real-time data, which, when analyzed, offers unprecedented operational insights and foresight. Cloud computing supports these endeavors by providing the computational and storage resources necessary to process and analyze vast time series datasets, enabling scalability and agility within industrial systems.

Despite its transformative influence, the integration of time series analysis within Industry 4.0 is not without challenges. The intricacies of deciphering intricate data patterns demand advanced analytical capabilities, while the integration of IoT introduces complexities in data security and system interoperability. Cloud computing must navigate concerns around data sovereignty and latency to maintain the integrity and responsiveness of time series analyses.

As we gaze into the future, the trajectory of Industry 4.0 will be significantly influenced by advancements in time series prediction. The growing sophistication of IoT systems promises richer, more granular data, enabling more nuanced and predictive analytics. Advancements in AI are poised to refine the accuracy of time series forecasting, enhancing the anticipatory capabilities of industrial systems. However, alongside these technological strides, the imperative of safeguarding data security and privacy becomes more acute, necessitating robust strategies to protect the integrity of time series data.

The path ahead also invites a multidisciplinary approach to crisis management, building upon the lessons learned during the pandemic. The intersection of medical knowledge, industrial pragmatism, and academic innovation is expected to yield robust solutions for future societal challenges. Moreover, the principles of Industry 4.0, when applied to urban development, hold the potential to transform cities and communities, infusing them with smart, sustainable, and interconnected systems driven by predictive analytics [105].

In sum, the reviewed literature presents a rich tapestry of insights into the vast potential of time series analysis as a cornerstone of Industry 4.0. The judicious combination of cutting-edge technologies that revolve around time series prediction is poised to revolutionize decision-making processes, refine industrial operations, and enhance crisis management. As we continue to embrace and advance the principles of Industry 4.0, the horizon is bright with the promise of a more efficient, resilient, and insightful future, underpinned by the profound capabilities of time series analysis.

**Author Contributions:** Conceptualization, J.B. and N.K.; investigation, N.K. and C.P.-I.; resources, J.B.; writing—original draft preparation, N.K.; writing—review and editing, J.B. and C.P.-I.; visualization, N.K.; supervision, J.B.; project administration, J.B.; funding acquisition, J.B. All authors have read and agreed to the published version of the manuscript.

**Funding:** This research was funded by the Polish National Science Centre project "Process Fault Prediction and Detection", contract no. UMO-2021/41/B/ST7/03851.

**Institutional Review Board Statement:** Not applicable.

**Informed Consent Statement:** Not applicable.

**Data Availability Statement:** No new data were created or analyzed in this study. Data sharing is not applicable to this article.

**Conflicts of Interest:** The authors declare no conflict of interest.

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
