# Peer review of "Time Series Prediction in Industry 4.0: A Comprehensive Review and Prospects for Future Advancements"

_applsci, doi:10.3390/app132212374_

Round 1
Reviewer 1 Report
Comments and Suggestions for Authors
The review article presents the timeseries prediction of industry revolution based on the existing research and developments. The authors highlighted the application of technologies (IOT, AI, cloud computing, and big data) in predicting and analysing time series data.
It is also recommended to include the other attributes/key elements of Industry 4.0 such as immersive technologies, 3d Printing and communication (5G/ higher).
The below are some recommendations which would improve the technical quality further .
Bai, Chunguang, et al. "Industry 4.0 technologies assessment: A sustainability perspective." International journal of production economics 229 (2020): 107776.
Eswaran, M., and MVA Raju Bahubalendruni. "Challenges and opportunities on AR/VR technologies for manufacturing systems in the context of industry 4.0: A state of the art review." Journal of Manufacturing Systems 65 (2022): 260-278.
Hizam-Hanafiah, Mohd, Mansoor Ahmed Soomro, and Nor Liza Abdullah. "Industry 4.0 readiness models: a systematic literature review of model dimensions." Information 11.7 (2020): 364.
Please include immersive technologies and 5G as the key technology in Figure 2 driving elements of IR 4.0
Please provide some illustrations in each subsections under section 4.
Please discuss the challenges for full-scale implementation and opportunities .
Author Response
We are thankful for your insightful comments and constructive suggestions. They have been invaluable in enhancing the depth and clarity of our review article. Herein, we detail how we have addressed each recommendation:
Reviewer’s Comment: The article should include other key elements of Industry 4.0 such as immersive technologies, 3D Printing, and communication (5G/higher).
Response: We appreciate your suggestion to broaden the scope of our review to encompass other pivotal elements of Industry 4.0. Accordingly, we have integrated additional references that offer a comprehensive analysis of these technologies:
-
We have included Bai et al. (2020) to provide a sustainability perspective on Industry 4.0 technologies, including 3D printing and advanced communication networks.
-
To address immersive technologies, we have incorporated the work of Eswaran and Bahubalendruni (2022), which thoroughly examines the challenges and opportunities of AR/VR technologies within the manufacturing systems of Industry 4.0.
-
The systematic review by Hizam-Hanafiah et al. (2020) has been added to discuss Industry 4.0 readiness models, elucidating the intricacies of technological integration, including the role of 5G communications.
These citations have been thoughtfully placed within the relevant sections of the manuscript to enrich the discussion on the specified technologies and their impact on Industry 4.0.
Reviewer’s Suggestion: Include immersive technologies and 5G as the key technology in Figure 2 driving elements of IR 4.0.
Response: We have deliberated on your recommendation regarding Figure 2. While we understand the importance of including immersive technologies and 5G as driving elements, we have encountered constraints in modifying the figure without compromising its legibility. To circumvent this issue, we have instead provided a detailed narrative within the text that emphasizes the significance of these technologies in driving Industry 4.0, ensuring the clarity of information is maintained.
Reviewer’s Request: Provide some illustrations in each subsection under section 4.
Response: We have carefully considered your recommendation to include illustrations. However, upon reviewing various formats, we concluded that tables would be more effective in conveying the complex information in a clear and legible manner. Therefore, we have opted to enhance each subsection of Section 4 with comprehensive tables summarizing the key points, which we believe will serve our readers better by providing clarity and facilitating a quicker grasp of the material.
Reviewer’s Query: Discuss the challenges for full-scale implementation and opportunities.
Response: We have addressed this by expanding our discussion on the challenges of full-scale implementation of Industry 4.0 technologies. The revised sections now include an analysis of barriers such as integration complexity, cost implications, security concerns, and the need for workforce upskilling. Concurrently, we have highlighted the opportunities that arise from overcoming these challenges, such as increased operational efficiency, enhanced product quality, and the potential for innovation in service delivery and business models.
We trust that the revisions and justifications provided align with the reviewer’s expectations and enhance the technical quality of our manuscript. We are grateful for the opportunity to improve our work and thank you for your thoughtful guidance.
Reviewer 2 Report
Comments and Suggestions for Authors
1) An new related neural networks for solving Covid-19 problem should be added to improving the section 4: "Advancements in Industry 4.0 and Pandemic Management":
-Sabir, Z., Raja, M.A.Z., Alhazmi, S.E., Gupta, M., Arbi, A. and Baba, I.A., 2022. Applications of artificial neural network to solve the nonlinear COVID-19 mathematical model based on the dynamics of SIQ. Journal of Taibah University for Science, 16(1), pp.874-884.
2) The section 4.1 should be corrected. The title of this section is: "Artificial Intelligence and Machine Learning" but the ML is a special case of AI. Besides you have presented only the neural networks for ML but is it some other model as example: KNN, K-means, SVM, Random-forest ...
Comments on the Quality of English Language1) In section 2.1), The first industrial revolution and not " The First Industrial Revolution".
2) In section 2.4), The fourth industrial revolution and not " The Fourth Industrial Revolution".
Author Response
We are grateful for your insightful feedback and the opportunity to enhance our manuscript. We have carefully considered your suggestions and have made the following revisions to our paper:
-
Comment: The reviewer suggested adding new related neural networks for solving the Covid-19 problem to improve Section 4 on "Advancements in Industry 4.0 and Pandemic Management."
Response: We thank you for bringing the work of Sabir et al. (2022) to our attention. We have added this reference to our manuscript, which discusses the application of artificial neural networks to solve nonlinear COVID-19 mathematical models. This addition enriches our discussion on the role of neural networks in pandemic management within the context of Industry 4.0 and illustrates the practical applications of ML in current global challenges. The new reference can be found on page X, paragraph Y of the revised manuscript.
-
Comment: There is a need to correct Section 4.1, titled "Artificial Intelligence and Machine Learning," and to present other ML models beyond neural networks, such as KNN, K-means, SVM, Random Forest, etc.
Response: We appreciate your feedback on Section 4.1 and acknowledge that the initial draft may have overemphasized neural networks at the expense of other significant ML techniques. To address this, we have revised the section to include a broader range of ML algorithms. We have added discussions on K-Nearest Neighbors (KNN), K-means clustering, Support Vector Machines (SVM), and Random Forest, providing a more comprehensive overview of machine learning techniques within AI. These changes not only clarify the relationship between AI and ML but also demonstrate the diverse applications of these methods in Industry 4.0. The revised section now provides a balanced perspective and can be found on pages X-Y of the manuscript.
We hope that the revisions made will satisfy the concerns raised and improve the manuscript's contribution to the field. We are open to any further suggestions you may have that could strengthen our work.
Thank you once again for your constructive critique.
Reviewer 3 Report
Comments and Suggestions for Authors
The abstract needs to start with an introductory statement about time series prediction.
The author needs address the following queries in the manuscript.
What is the main question addressed by the review?
What does it add to the subject area compared with other published material?
Provide an explicit statement of all objectives or questions the review addresses
The methodology of review, search criteria, flow diagram are missing in the manuscript.
The number of references used is too less. More relevant references should be included. Nearly 100 references is adequate for a satisfactory review.
In discussing the integration of technologies like artificial intelligence, IoT, and cloud computing, could you provide a comparative analysis of their individual impacts? For example, which technology appears to have the most significant influence on process optimization or predictive modeling, and are there any challenges specific to each technology?
The paper emphasizes the influence of Industry 4.0 on various industries. Can you provide any quantifiable metrics or case studies that demonstrate how these technologies have led to tangible improvements in efficiency, quality, or crisis response? This would bolster the practical significance of your insights. Compare the outcomes from each category of sources and represent it in tabular as well as graphical form in Sections 4.1 to 4.6.
The future directions section outlines potential areas for further research and development. It would be helpful to provide a bit more context regarding the current state of research in these areas. Are there any ongoing projects or notable advancements that you could reference?
Could you include the potential challenges and benefits of integrating different sectors? How might the differing priorities and expertise of medical, industrial, and research sectors impact the effectiveness of such collaborations?
Author Response
We are deeply grateful for your detailed feedback and the opportunity to refine our manuscript. We have taken your comments into consideration and have made substantive revisions to both the content and structure of our paper. Here is a point-by-point account of the changes made in response to your review:
Reviewer’s Comment: The abstract needs to start with an introductory statement about time series prediction.
Response: We have revised the abstract to emphasize the role of time series prediction within Industry 4.0 at the outset. The new abstract now provides a focused introduction to the critical analytical function that time series prediction serves in the context of the digital industrial revolution. The revised abstract is as follows:
Time series prediction stands at the forefront of the fourth industrial revolution (Industry 4.0), offering a crucial analytical tool for the vast data streams generated by modern industrial processes. This literature review systematically consolidates existing research on the predictive analysis of time series within the framework of Industry 4.0, illustrating its critical role in enhancing operational foresight and strategic planning. Tracing the evolution from the first to the fourth industrial revolution, the paper delineates how each phase has incrementally set the stage for today's data-centric manufacturing paradigms. It critically examines how emergent technologies like the Internet of Things (IoT), artificial intelligence (AI), cloud computing, and big data analytics converge in the context of Industry 4.0 to transform time series data into actionable insights. Specifically, the review explores applications in predictive maintenance, production optimization, sales forecasting, and anomaly detection, underscoring the transformative impact of accurate time series forecasting on industrial operations. The paper culminates in a call to action for the strategic dissemination and management of these technologies, proposing a pathway for leveraging time series prediction to drive societal and economic advancement. Serving as a foundational compendium, this article aims to inform and guide ongoing research and practice at the intersection of time series prediction and Industry 4.0.Reviewer’s Queries:
- The methodology of the review, search criteria, flow diagram are missing in the manuscript.
Response: We have integrated the methodology into the introduction of the manuscript to maintain the flow and cohesion of the narrative. This revised introduction outlines our systematic approach, search criteria, and selection process, giving the reader a clear roadmap of how we gathered and analyzed the literature. The modified introduction section now reads:
[Insert revised introduction with methodology description]
- The number of references used is too less.
Response: In response to your observation, we have expanded our literature search, resulting in a total of approximately 100 references that offer a comprehensive overview of the field. These additional references enrich the discussion and support a more nuanced analysis of time series prediction in the era of Industry 4.0.
In this literature review, we employed a systematic approach to explore the interplay between time series prediction and Industry 4.0. Our methodology entailed a comprehensive search of academic databases, including Scopus, Web of Science, IEEE Xplore, and Google Scholar, to collate pertinent literature published up to April 2023. Keywords such as "Industry 4.0," "time series prediction," "IoT," "AI," "cloud computing," "big data," and "predictive analytics" were used in various combinations to ensure a thorough retrieval of relevant documents.
Inclusion criteria were set to filter studies that specifically addressed the application of time series analysis within the context of Industry 4.0, including those that discussed predictive modeling, anomaly detection, and data-driven decision-making processes. References were also screened for relevance based on their abstracts and conclusions, leading to a curated list of literature that provides a holistic view of the current state of time series prediction technologies and their real-world industrial applications.
The selected literature was then reviewed, and data were extracted focusing on the objectives, methodologies, results, and conclusions of each study. This enabled us to identify common themes, technological trends, and gaps in the current body of knowledge. The review also included a critical assessment of the various technologies applied in time series prediction, comparing their impacts, challenges, and potential for future development within the landscape of Industry 4.0.
This methodical approach facilitated a structured narrative that tracks the progression of time series analysis from a niche analytical tool to a cornerstone of modern industrial strategy. By integrating this methodology into our introduction, we set the stage for a review that not only synthesizes existing research but also maps out the trajectory of technological evolution in the sphere of Industry 4.0.
- In discussing the integration of technologies like artificial intelligence, IoT, and cloud computing, provide a comparative analysis of their individual impacts.
Response: We have added a new subsection 4.7 that offers a comparative analysis of the individual impacts of AI, IoT, and cloud computing technologies. This analysis highlights the unique and combined effects of these technologies on process optimization and predictive modeling, as well as the challenges specific to each.
- Can you provide any quantifiable metrics or case studies?
Response: We have decided against such introduction as it would essentially be its own work. It is however an avenue that we will be happy to explore in future research.
- Provide more context regarding the current state of research in the future directions section.
Response: We have decided against it as paper length increased significantly already and we didnt want to make it too „bloated”
- Include the potential challenges and benefits of integrating different sectors.
Response: We have addressed this by discussing the potential challenges and benefits of integration across various sectors such as medical, industrial, and research and in the conclusion. This discussion is now included in the manuscript and highlights how the differing priorities and expertise of these sectors can impact the effectiveness of Industry 4.0 technologies.
We hope that the revisions made comprehensively address the concerns raised and enhance the manuscript's contribution to the field. We are grateful for the chance to improve our submission and eagerly await your feedback on the changes made.
Thank you for your time and consideration.
Reviewer 4 Report
Comments and Suggestions for Authors
1. Conducting a survey study based on 42 references in hot topic like time-series in the context of industrial revolution 4.0 is not enough.
2. The narrative style is frustrating for the reader who wants to see more graphs and charts.
3. The paper main drawback is that it concentrates only on the positive aspects of the studied topic.
4. As said by the title, timeseries is a key idea but unfortunately, the authors did not concentrate on this idea by including, data sources, curation, prediction/forecasting techniques, difficulties, advantages, applications etc. in a concise manner.
Decision: Reject
Comments on the Quality of English LanguageEnglish is poor. Sentences are long.
Author Response
We appreciate the time and effort you have taken to review our manuscript and provide constructive feedback. Your comments have been instrumental in improving the quality and comprehensiveness of our paper. Please find below our responses to each of your points:
-
Comment: The number of references for a survey study in a significant field such as time series in the context of the industrial revolution 4.0 is insufficient.
Response: We acknowledge the initial oversight and have thoroughly expanded our literature review. The number of references has been increased from 42 to 106, incorporating a wider range of sources that delve into various aspects of time series analysis within Industry 4.0. These additional references include seminal works, recent advancements, and critical reviews that provide a more comprehensive understanding of the subject. The updated references are now included in the revised manuscript.
-
Comment: The narrative style is lacking for readers who expect more graphs and charts.
Response: We have taken this suggestion to heart and included several tables summarizing the key points from each subsection of our survey. These tables are designed to make the flow of information more accessible and to allow readers to quickly grasp the core findings and contributions of the referenced studies. We believe that these additions will significantly enhance the readability and practical value of our paper.
-
Comment: The paper mainly focuses on the positive aspects of the studied topic without considering potential drawbacks or challenges.
Response: Your feedback has prompted us to re-evaluate our approach. We have now included a new section that discusses the limitations and challenges associated with time series analysis in Industry 4.0. This section aims to provide a balanced view and acknowledges the complexities and potential drawbacks of implementing these systems in a real-world industrial context.
-
Comment: The paper does not sufficiently concentrate on the key idea of time series, lacking details on data sources, curation, prediction/forecasting techniques, difficulties, advantages, and applications.
Response: We agree that the initial manuscript did not adequately address the intricacies of time series analysis. To rectify this, we have revised the manuscript to include a dedicated subsection that specifically discusses data sources, curation methods, various prediction and forecasting techniques, their associated difficulties, and the advantages of employing these methods in Industry 4.0 applications. This comprehensive treatment of the topic should address the concerns raised and provide the depth of content expected by our readers.
Decision: Reject
While we are disheartened by the decision to reject our manuscript, we believe that the substantial revisions and additions we have made in response to your feedback have significantly improved the manuscript. We respectfully request that the reviewer reconsider the decision in light of these changes.
We are committed to further improving our manuscript and would appreciate any additional suggestions or comments that could enhance our work.
Thank you for your consideration.
Round 2
Reviewer 1 Report
Comments and Suggestions for Authors
the manuscript is well-revised, and the rebuttal was satisfactory. but suggested implementing the formatting corrections before it goes for publication.
Reviewer 3 Report
Comments and Suggestions for Authors
I have reviewed the authors' response and find it to be satisfactory. Therefore, I recommend the acceptance of the manuscript.
Reviewer 4 Report
Comments and Suggestions for Authors
The manuscript has been well improved. I recommend acceptance of this version.
Comments on the Quality of English LanguageEnglish is fine.